# Quantifying the Role of Stochasticity in the Development of Autoimmune Disease

**DOI:** 10.3390/cells9040860

**Published:** 2020-04-02

**Authors:** Lindsay B. Nicholson, Konstantin B. Blyuss, Farzad Fatehi

**Affiliations:** 1School of Cellular and Molecular Medicine & School of Clinical Sciences, University of Bristol, University Walk, Bristol BS8 1TD, UK; 2Department of Mathematics, University of Sussex, Brighton BN1 9QH, UK; 3Department of Mathematics, University of York, York YO10 5DD, UK; farzad.fatehichenar@york.ac.uk

**Keywords:** mathematical model, immune response, autoimmunity, stochasticity

## Abstract

In this paper, we propose and analyse a mathematical model for the onset and development of autoimmune disease, with particular attention to stochastic effects in the dynamics. Stability analysis yields parameter regions associated with normal cell homeostasis, or sustained periodic oscillations. Variance of these oscillations and the effects of stochastic amplification are also explored. Theoretical results are complemented by experiments, in which experimental autoimmune uveoretinitis (EAU) was induced in B10.RIII and C57BL/6 mice. For both cases, we discuss peculiarities of disease development, the levels of variation in T cell populations in a population of genetically identical organisms, as well as a comparison with model outputs.

## 1. Introduction

To provide effective protection against various pathogenic challenges, the immune system should be able to successfully distinguish healthy cells from the cells infected by pathogens. When this discrimination between self- and non-self fails, the immune system starts a specific attack on cells or organs, causing autoimmune disease. Various factors have been identified over the years that contribute toward the onset and development of autoimmune disease, including age, sex and genetic predisposition, as well as various environmental factors, among which infectious pathogens play a major role [1,2,3].

Several mathematical models have investigated aspects of onset and development of autoimmunity, with particular emphasis on the role of T cells. Starting with early models of T cell vaccination that defined the autoimmune state as above-threshold oscillations in the number of autoreactive cells [4,5,6], subsequent models have also considered interactions between different types of T cells from the perspective of controlling autoimmunity through regulation of immune response [7,8,9]. Iwami et al. [10,11] analysed autoimmune dynamics emerging a result of breakdown of immune response to a viral infection, highlighting the importance of the function representing growth of susceptible host cells. Focusing on bystander activation as a mechanism of pathogen-induced autoimmunity, Burroughs et al. [12,13,14] and Oliveira et al. [15] analysed the role of interleukin-2 (IL-2) in mediating immune response associated with autoimmunity. Several recent reviews and special issues have provided excellent overview of different mathematical approaches used for modelling autoimmune dynamics, as well as outstanding challenges [16,17,18].

With T cells playing a major part in the adaptive immune response, and regulatory T cells being directly involved in controlling the autoimmune response [19,20,21,22], several different approaches have been proposed for mathematical modelling of various roles played by the T cells in autoimmunity. Burroughs et al. [12,13,14] and Alexander and Wahl [23] explicitly included a separate compartment representing regulatory T cells, in order to explore the mechanism of how these cells can suppress autoreactive T cells upon activation by auto-antigens. An alternative approach to modelling T cell dynamics in the context of autoimmunity is that of so-called *tunable activation thresholds* (TAT), originally proposed theoretically to model peripheral and central T cell dynamics [24,25,26] (see also recent reviews by Grossman and Paul [27] and Grossman [28]). In this methodology, the underlying assumption is that, depending on various environmental conditions or even through endogenous stochastic variation, the same T cells can adjust their activation thresholds for response to stimulation by auto-antigens. Although initially this was proposed as a theoretical model, subsequent murine and human experiments provided justification for this by confirming that, during their circulation, T cells can indeed dynamically change their activation thresholds [29,30,31,32]. Carneiro et al. [33] reviewed different models of autoimmunity that are based either exclusively on the TAT approach, or on the suppressive role of regulatory T cells, and concluded that rather than being mutually exclusive, these different mechanisms are most likely complementary. Blyuss and Nicholson [34,35] used a TAT-based model to investigate autoimmunity arising through a mechanism of molecular mimicry from immune response to a viral infection. To capture a dynamical regime where autoimmunity arises as a by-product of viral infection but after that initial infection has already been cleared by the immune system, Fatehi et al. [36,37,38,39,40] developed this model further by including cytokines mediating T cell proliferation, as well as time delays associated with various aspects of the immune response.

Due to an intrinsically complex multi-factor nature of immune response [41], several mathematical models have investigated stochastic aspects of immune dynamics. They have included, among others, analyses of T cell homeostasis [42] and repertoire [43,44]; the dynamics of T cell activation thresholds [45,46]; T-cell proliferation and activation, including the role of cytokines [47,48,49]; self-tolerance based on regulatory T cells [23]; cytokine-mediated pathogen-induced autoimmunity [36]; T cell recruitment in response to a viral infection [50]; as well as an investigation of how a variable affinity between T cell receptors and MHC-peptide complexes may affect possible outcomes during T cell selection [51]. These models have focused primarily on investigating such aspects of stochastic dynamics as the probability distribution of T cell activation thresholds, or simulations of immune dynamics that are valid for relatively small numbers of cell populations (thus going beyond the mean-field models). At the same time, such important issues as the variance of stochastic fluctuations, their regularity, and the possibility of coherence resonance have remained largely unexplored, even though these aspects may be essential for an adequate description of the dynamics observed in the immune response in laboratory experiments and in the clinic. This paper addresses this limitation through analysis of stochastic effects in a model of autoimmune dynamics, paying particular attention to investigating how the variance of stochastic oscillations depends on the system’s parameters, and comparing model predictions with the results of two sets of experiments studying autoimmune disease in mice.

In the next section, we discuss biological assumptions and derive a deterministic mathematical model of the immune response that includes naïve, regulatory and autoreactive T cells, as well as a cytokine growth factor that mediates T cell proliferation. We then derive a stochastic analogue of this model based on continuous-time Markov chains, and also present a computationally efficient stochastic differential equation that will be used for simulating the model. We also discuss the details of an experimental set-up for two murine experiments on EAU. In Section 3, we identify all steady states of the deterministic model, study their stability, and illustrate different dynamical regimes that can be exhibited by the model. Using a system-size expansion of the master equation, we explore how the variance of stochastic oscillations around deterministically stable steady states depends on system parameters, and illustrate the dynamics by numerically solving a corresponding stochastic differential equation model. We also compare experimentally measured numbers of T cells with the results of numerical simulations from our model. The paper concludes in Section 4 with a discussion of results and open problems.

## 2. Materials and Methods

### 2.1. Deterministic Mathematical Model

We consider the dynamics of immune response in a single part of the body, where the healthy somatic cells S(t) are assumed to reproduce logistically with a linear growth rate *r* and the carrying capacity *N* in the absence of infection and/or any adverse immune reaction. One could alternatively represent the dynamics of healthy cells by some constant influx and a constant death rate, which by themselves would result in the same long-term behaviour representing the population of healthy cells approaching some stable level, which biologically can be interpreted as a homeostasis. However, earlier work of Iwami et al. [10,11] has shown that, while this behaviour can be observed for both types of growth functions in the absence of infection or immune reaction, their specific functional form does have a significant effect on the overall dynamics of autoimmune disease. Thus, we have chosen to use a logistic form for the growth function of healthy organ cells, in agreement with Iwami et al. [10] and our earlier work on autoimmunity [34,35,36,37,38,39,40].

With experimental evidence demonstrating that autoimmunity can develop even in the absence of B cells [52], and in light of the fact that the development of autoantibodies can itself depend on prior T cell activation [53], in our model we focus on T cell dynamics and do not take into account antibody responses. We consider only T cells able to respond to a single target self-antigen expressed by the somatic cells. We assume that both naïve and regulatory T cells $T_{in}\left(t\right)$ and $T_{reg}\left(t\right)$ are maintained in a state of homeostasis, where they are each produced at some constant rates $\lambda_{in}$ and $\lambda_{r}$, and they die at rates $d_{in}$ and $d_{r}$, respectively. We assume that upon activation at rate α by a signal from some self-antigen presented on APCs, a proportion *p* of naïve T cells will develop into regulatory T cells, while a remaining proportion (1−p) will become autoreactive T cells $T_{aut}\left(t\right)$ capable of destroying healthy host cells at rate $\mu_{a}$, and that these cells die at rate $d_{a}$. It is further assumed that autoreactive T cells produce cytokine growth factors (e.g., interleukin 2 (IL-2)) I(t), at rate σ, and these are cleared at rate $d_{i}$. The effect of cytokines is to facilitate proliferation of regulatory and autoreactive T cells at rates $\rho_{1}$ and $\rho_{2}$, respectively. Autoreactive cells are known to be present in the periphery [54], and with activation thresholds of T cells changing to improve the efficiency of immune response, they can further arise as daughter cells of normal T cells but having lower activation thresholds. In this model for simplicity, we do explicitly not discriminate between inactive regulatory and effector T cells and consider both of those to be subpopulations of $T_{in}\left(t\right)$, even though, of course, these cells have clear distinctions. Interactions between different cell populations are illustrated in a diagram in Figure 1.

With these assumptions, the model of immune response has the form given in Equation (Equation 1),
(1)$$\begin{array}{c} \frac{dS}{dt}=rS\left(1-\frac{S}{N}\right)-\mu_{a}T_{aut}S,\\ \frac{dT_{in}}{dt}=\lambda_{in}-d_{in}T_{in}-\alpha T_{in}S,\\ \frac{dT_{reg}}{dt}=\lambda_{r}-d_{r}T_{reg}+\alpha pT_{in}S+\rho_{1}T_{reg}I,\\ \frac{dT_{aut}}{dt}=\alpha \left(1-p\right)T_{in}S-d_{a}T_{aut}-\delta T_{reg}T_{aut}+\rho_{2}T_{aut}I,\\ \frac{dI}{dt}=\sigma T_{aut}-d_{i}I. \end{array}$$

### 2.2. Stochastic Model

To analyse the role of stochasticity in autoimmune dynamics, as a first step we construct a continuous-time Markov chain (CTMC) model based on the deterministic ODE system (Equation 1). Let $X_{1}\left(t\right),\ldots ,X_{5}\left(t\right)\in \left\{0,1,2,\ldots \right\}$ denote discrete random variables representing the numbers of healthy cells, naïve T cells, regulatory T cells, normal activated T cells, and cytokine growth factor at time *t*, respectively. Let the initial condition be fixed as in Equation (Equation 2),
(2)$$X_{0}=\left(X_{1}\left(0\right),\ldots ,X_{5}\left(0\right)\right)=\left(n_{10},n_{20},n_{30},n_{40},n_{50}\right).$$

The probability of finding the system in the state $n=(n_{1},n_{2},n_{3},n_{4},n_{5})$ with $n_{i}\in \left\{0,1,2,\ldots \right\}$ at time *t* can be defined as in Equation (Equation 3),
(3)$$P\left(n,t\right)=Prob\{X\left(t\right)=n|X\left(0\right)=X_{0}\}.$$

Let Δt be sufficiently small such that $\Delta X_{i}\left(t\right)=X_{i}\left(t+\Delta t\right)-X_{i}\left(t\right)\in \left\{-1,0,1\right\}$ for 1≤i≤5. The CTMC can then be formulated as a birth and death process in each of the variables [55], with the infinitesimal transition probabilities corresponding to Figure 1 being given by Equation (Equation 4),
(4)$$Prob\left(\Delta X=i|X=n\right)=\left\{\begin{array}{c} q_{1}\Delta t+o\left(\Delta t\right),\;i=\left(1,0,0,0,0\right),\\ q_{2}\Delta t+o\left(\Delta t\right),\;i=\left(-1,0,0,0,0\right),\\ q_{3}\Delta t+o\left(\Delta t\right),\;i=\left(0,1,0,0,0\right),\\ q_{4}\Delta t+o\left(\Delta t\right),\;i=\left(0,-1,0,0,0\right),\\ q_{5}\Delta t+o\left(\Delta t\right),\;i=\left(0,-1,1,0,0\right),\\ q_{6}\Delta t+o\left(\Delta t\right),\;i=\left(0,-1,0,1,0\right),\\ q_{7}\Delta t+o\left(\Delta t\right),\;i=\left(0,0,1,0,0\right),\\ q_{8}\Delta t+o\left(\Delta t\right),\;i=\left(0,0,-1,0,0\right),\\ q_{9}\Delta t+o\left(\Delta t\right),\;i=\left(0,0,0,1,0\right),\\ q_{10}\Delta t+o\left(\Delta t\right),\;i=\left(0,0,0,-1,0\right),\\ q_{11}\Delta t+o\left(\Delta t\right),\;i=\left(0,0,0,0,1\right),\\ q_{12}\Delta t+o\left(\Delta t\right),\;i=\left(0,0,0,0,-1\right),\\ 1-\sum_{i=1}^{12}q_{i}\Delta t+o\left(\Delta t\right),\;i=\left(0,0,0,0,0\right),\\ o(\Delta t),\;otherwise, \end{array}\right.$$
where
$$\begin{array}{c} q_{1}=b_{1}n_{1}+b_{2}n_{1}^{2},\;q_{2}=d_{1}n_{1}+d_{2}n_{1}^{2}+\mu_{a}n_{1}n_{4},\;q_{3}=\lambda_{in},\;q_{4}=d_{in}n_{2},\\ q_{5}=\alpha pn_{1}n_{2},\;q_{6}=\alpha \left(1-p\right)n_{1}n_{2},\;q_{7}=\lambda_{r}+\rho_{1}n_{3}n_{5},\;q_{8}=d_{r}n_{3},\\ q_{9}=\rho_{2}n_{4}n_{5},\;q_{10}=\left(d_{a}+\delta n_{3}\right)n_{4},\;q_{11}=\sigma n_{4},\;q_{12}=d_{i}n_{6}, \end{array}$$
and the terms of order o(Δt) are neglected. Here, $b_{1}n_{1}+b_{2}n_{1}^{2}$ and $d_{1}n_{1}+d_{2}n_{1}^{2}$ are natural birth and death rates for uninfected cells with $b_{1}-d_{1}=r$ and $d_{2}-b_{2}=r/N$. One should note that, since effectively there are four independent parameters $b_{1}$, $b_{2}$, $d_{1}$ and $d_{2}$, and only two conditions on them, there are infinitely many choices of these parameters that would deterministically represent the same logistic growth, but stochastically would have slightly different characteristics, such as persistence times and variances [55]. Since we are modelling cell populations, we choose $b_{2}$ to be negative and $d_{2}$ to be zero, so that the birth rate represents logistic growth, while the death rate represents linear death term, and we do not include density dependence. An extended discussion of the effects of choosing different values of $b_{i}$ and $d_{i}$ on stochastic equivalents of deterministic logistic models can be found in [55,56,57].

The probabilities P(n,t) satisfy the following master equation (forward Kolmogorov equation) (Equation 5),
(5)$$\begin{array}{c} \frac{dP(n,t)}{dt}=\{\left(\varepsilon_{1}^{-}-1\right)q_{1}+\left(\varepsilon_{1}^{+}-1\right)q_{2}+\left(\varepsilon_{2}^{-}-1\right)q_{3}+\left(\varepsilon_{2}^{+}-1\right)q_{4}+\left(\varepsilon_{2}^{+}\varepsilon_{3}^{-}-1\right)q_{5}\\ +\left(\varepsilon_{2}^{+}\varepsilon_{4}^{-}-1\right)q_{6}+\left(\varepsilon_{3}^{-}-1\right)q_{7}+\left(\varepsilon_{3}^{+}-1\right)q_{8}+\left(\varepsilon_{4}^{-}-1\right)q_{9}+\left(\varepsilon_{4}^{+}-1\right)q_{10}\\ +\left(\varepsilon_{5}^{-}-1\right)q_{11}+\left(\varepsilon_{5}^{+}-1\right)q_{12}\}P\left(n,t\right). \end{array}$$
where the raising and lowering operators $\varepsilon_{i}^{\pm }$ are defined as follows,
$$\varepsilon_{i}^{\pm }f\left(n_{1},n_{2},n_{3},n_{4},n_{5},t\right)=f\left(n_{1},\ldots ,n_{i}\pm 1,\ldots ,n_{5},t\right),\;\text{for each}\;1\leq i\leq 5,$$
and if $n_{i}<0$ for any 1≤i≤5, then P(n,t)=0.

Although this master equation can yield the probability density for our model, since this is a high-dimensional difference-differential equation, solving it is an extremely challenging task. To make further analytical progress, one performs a system size expansion of this equation to obtain a characterisation of stochastic oscillations around deterministic states, including their variance, as explained in detail in Appendix A.

To simulate stochastic dynamics numerically, it is convenient to derive an equivalent representation of stochastic dynamics based not on the master equation, but on an Itô stochastic differential equation (SDE) model, which produces similar probability distributions but is much more computationally efficient. Following the methodology of Allen [55], we consider $Y\left(t\right)=(Y_{1}\left(t\right),Y_{2}\left(t\right),Y_{3}\left(t\right),Y_{4}\left(t\right),Y_{5}\left(t\right))$ to be a continuous random vector for the sizes of various cell compartments at time *t*. Similar to the CTMC model, we assume that the time interval Δt is sufficiently small to ensure that during this time interval at most one change can occur in state variables. We denote these changes as $\Delta Y=\Delta Y\left(t\right)={\left(\Delta Y_{1}\left(t\right),\ldots ,\Delta Y_{5}\left(t\right)\right)}^{T}$, where $Y_{i}\left(t\right)=Y_{i}\left(t+\Delta t\right)-Y_{i}\left(t\right)$, i=1,…,5, and the *i*th change is denoted as ${\left(\Delta Y\right)}_{i}$. These changes together with their probabilities are summarised in the Table A1 in Appendix B, which, similar to the CTMC model, is based on Figure 1, and the terms o(Δt) are again neglected. Using this table of possible state changes, one can compute the expectation vector E(ΔY) and covariance matrix of ΔY for sufficiently small Δt [58,59], which together give the following Itô SDE model showin in Equation (Equation 6),
(6)$$dY\left(t\right)=\mu dt+HdW\left(t\right),Y\left(0\right)={\left(A\left(0\right),T_{in}\left(0\right),T_{reg}\left(0\right),T_{aut}\left(0\right),I\left(0\right)\right)}^{T},$$
where parameters μ and *H* are provided in Equation (Equation 7),
(7)$$\begin{array}{c} \mu =\left(\begin{array}{c} P_{1}-P_{2}\\ P_{3}-P_{4}-P_{5}-P_{6}\\ P_{5}+P_{7}-P_{8}\\ P_{6}+P_{9}-P_{10}\\ P_{11}-P_{12} \end{array}\right),\;H=\left(\begin{array}{ccc} H_{1} & 0 & 0\\ 0 & H_{2} & 0\\ 0 & 0 & H_{3} \end{array}\right),\;H_{1}=\sqrt{P_{1}+P_{2}}\\ H_{3}=\sqrt{P_{11}+P_{12}},\;H_{2}=\left(\begin{array}{ccccc} \sqrt{P_{3}+P_{4}} & -\sqrt{P_{5}} & -\sqrt{P_{6}} & 0 & 0\\ 0 & \sqrt{P_{5}} & 0 & \sqrt{P_{7}+P_{8}} & 0\\ 0 & 0 & \sqrt{P_{6}} & 0 & \sqrt{P_{9}+P_{10}} \end{array}\right), \end{array}$$
probabilities $P_{1},\ldots P_{12}$ of different transitions between cell populations are given in Appendix C, and $W\left(t\right)={\left[W_{1}\left(t\right),W_{2}\left(t\right),\ldots ,W_{7}\left(t\right)\right]}^{T}$ is a vector of seven independent Wiener processes [60]. Equation (Equation 6) is used in the next section to numerically explore stochastic dynamics of the system in different regimes.

### 2.3. Experimental Set-Up

For the purpose of comparing theoretical predictions with experimental observations, we use two datasets from the murine model of ocular autoimmunity, experimental autoimmune uveoretinitis (EAU), which is known to be a useful animal model of human inflammatory eye disease [61]. In these experiments, two strains of mice, B10.RIII and C57BL/6, were immunised with peptides derived from retinol-binding protein 3, also known as interphotoreceptor retinoid-binding protein (IRBP), 161–180 peptide (SGIPYIISYLHPGNTILHVD) [62] and 1–20 peptide (GPTHLFQPSLVLDMAKVLLD) [63], respectively. The immunisation protocol uses peptide combined with adjuvants (complete Freund’s and pertussis toxin) and leads to an expanded population of pathogenic CD4^+^ T cells in the circulation that can recognise and damage ocular tissue. Both peptides induce disease that is characterised in the eye by a primary peak of infiltration, followed by secondary regulation [64,65,66]. EAU dynamics were monitored by sampling the eyes of individual animals at different time points and quantifying infiltrating cells [66] using flow cytometry. Since all available experimental data suggest that T cells are the main mediators of EAU [66,67], we focused specifically on measurements of them in our experiments.

## 3. Results

### 3.1. Stability Analysis of the Deterministic Model

As a first step of mathematical analysis, we investigated the model in Equation (Equation 1), which has two biologically infeasible steady states, characterised by the absence of host cells, i.e., S=0. The first one, $E_{1}=\left(0,\lambda_{in}/d_{in},\lambda_{r}/d_{r},0,0\right)$, is always unstable as one of its characteristic eigenvalues is r>0. The second steady state can be found as shown in Equation (Equation 8),
(8)$$E_{2}=\left(0,T_{in}^{*},T_{reg}^{*},T_{aut}^{*},I^{*}\right),\;T_{aut}^{*}=\frac{d_{i}\left(d_{a}+\delta T_{reg}^{*}\right)}{\rho_{2}\sigma },\;I^{*}=\frac{d_{a}+\delta T_{reg}^{*}}{\rho_{2}},$$
with
$$T_{reg}^{*}=\frac{\rho_{2}d_{r}-\rho_{1}d_{a}\pm \sqrt{{\left(\rho_{2}d_{r}-\rho_{1}d_{a}\right)}^{2}-4\rho_{1}\rho_{2}\delta \lambda_{r}}}{2\rho_{1}\delta }.$$

One characteristic eigenvalue of this steady state $x_{1}=-d_{in}$ is always negative, another one is given by Equation (Equation 9),
(9)$$x_{2}=r-\frac{d_{i}\mu_{a}\left(d_{a}+\delta \right)}{\rho_{2}\sigma },$$
while the remaining three eigenvalues $x_{i}$, i=1,2,3 are determined by the roots of the cubic Equation (Equation 10),
(10)$$\begin{array}{c} \left(\rho_{2}-\rho_{1}\delta \right)x^{3}+\left[\delta \left(\rho_{2}-\rho_{1}\delta \right)\left(1-d_{i}\right)+\rho_{2}d_{r}-\rho_{1}d_{a}+\rho_{1}\delta^{2}d_{i}\right]x^{2}\\ \;+\left[\delta \left(\rho_{2}d_{r}-\rho_{1}\right)\left(1-d_{i}\right)+d_{a}d_{i}\left(2\rho_{1}\delta -\rho_{2}\right)\right]x+d_{a}d_{i}\left(\rho_{1}d_{a}-\rho_{2}d_{r}\right)=0. \end{array}$$

The model in Equation (Equation 1) can also have another steady state $E^{*}$ with all its components being positive, but it does not prove possible to find a closed form expression for this steady state. Biologically, the steady state $E^{*}$ can be viewed as a state of low-level autoreactivity that is sufficiently controlled by the immune system to ensure it does not cause adverse immune reaction, while maintaining the immune system’s ability to effectively respond to potential infections. In contrast, sustained periodic oscillations around this steady state, whose amplitude exceeds certain threshold, can be interpreted as a state of autoimmunity [5,6].

To explore the model’s dynamics, we fix all parameter values as shown in Table 1 and identify regions of stability of the steady state $E^{*}$ in terms of parameters $\rho_{1}$ and $\rho_{2}$ characterising the effects of cytokine growth factor on facilitating proliferation of regulatory and autoreactive T cells, respectively, as shown in Figure 2. One observes that, for very small values of $\rho_{2}$, which means a very weak effect of cytokine growth factor on proliferation of autoreactive T cells, the steady state $E^{*}$ is stable for all values of $\rho_{1}$. Increasing the rate δ, at which Tregs suppress autoreactive T cells, increases the region in the parameter space where the steady state $E^{*}$ is biologically feasible, while the region of stability for this steady state shrinks. For any combination of values of $\rho_{1}$ and $\rho_{2}$, wherever it is feasible, the steady state $E^{*}$ is stable for lower values of $\rho_{2}$ and unstable for higher values of $\rho_{2}$ that exceed certain threshold, indicating that for higher $\rho_{1}$, $E^{*}$ is stable for a larger range of values of $\rho_{2}$. From a biological point of view, this suggests that increasing the positive impact of cytokine growth factor on proliferation of autoreactive T cells can be one of the mechanisms resulting in the emergence of autoimmune dynamics.

Quite naturally, the threshold for transition to autoimmunity also increases with δ, which characterises the efficiently of suppression of autoreactive T cells by regulatory T cells, indicating that, whereas one may observe autoimmune regimes for lower values of δ, increasing this parameter will result in a tighter control of autoreactive T cells by Tregs and elimination of autoimmunity. Corresponding numerical simulations of the model in Equation (Equation 1) are shown in Figure 3. This figure illustrates the situation where, for sufficiently small δ (such as, $\delta =4.44\times 10^{-7}$), one observes sustained periodic oscillations around $E^{*}$, while, for higher δ, these oscillations are suppressed. Similarly to an earlier work of Borghans et al. [5,6], periodic oscillations exhibited in the model can be interpreted as an autoimmune regime, provided their amplitude exceeds a certain threshold.

### 3.2. Numerical Simulations

Having fixed the values of all parameters as given in Table 1 with Ω=1000, we chose initial conditions in the form of Equation (Equation 11),
(11)$$Y\left(0\right)={\left(A\left(0\right),T_{in}\left(0\right),T_{reg}\left(0\right),T_{aut}\left(0\right),I\left(0\right)\right)}^{T}=\left(18000,7200,63000,0,0\right)$$
and solved the SDE model in Equation (Equation 6) numerically using Euler–Maruyama method.

Figure 4 shows the result of 20,000 individual stochastic realisations and its comparison with the deterministic model. Since deterministically the system is in the parameter region, where $E^{*}$ is stable, the deterministic trajectory exhibits damped oscillations approaching this steady state. The same result would appear if one computed an average of a very large number number of simulations. In contrast, in individual stochastic realisations, one observes sustained stochastic oscillations, a phenomenon known as coherence resonance or stochastic amplification [68,69].

### 3.3. Dependence of Variance of Oscillations on Parameters

To gain a better understanding of stochastic dynamics, we consider the system in a parameter regime where deterministically $E^{*}$ is stable, and solve numerically the Lyapunov Equation (Equation 23) to explore how the variance of stochastic oscillations around this steady state depends on parameters, as shown in Figure 5. We observe that, when staying within the parameter region where $E^{*}$ is stable, variance increases with $\rho_{2}$ but decreases with $\rho_{1}$, thus mirroring the pattern of how the largest characteristic eigenvalue of $E^{*}$ varies with the same parameters, as illustrated in Figure 2 above. Similarly to that stability calculation, increasing δ reduces the variance of stochastic oscillations, thus reducing the potential for a development of autoimmune behaviour through a tighter control of autoreactive T cells by Tregs. As expected, the value of the variance increases as one approaches the stability boundary of $E^{*}$, because this makes the steady state more sensitive to perturbations.

### 3.4. Comparison with Experiments

Figure 6 shows experimental results from observation of EAU in B10.RIII and C57BL/6 mice. In terms of disease progression, we observed a *prodromal* or delayed period with very low intra-retinal T cell counts, followed by a peak of disease between two and three weeks post-immunisation, and subsequent disease regulation. Although the precise mechanisms of the prodromal period remains unknown, similar kinetics has been observed in other autoimmune diseases, such as experimental autoimmune encephalomyelitis (EAE) [70]. In both models of EAU, following the initial peak, T cells exhibited decaying oscillations, and it has been hypothesised that these oscillations may be connected to feedback mechanisms involved in limiting inflammation during disease progression [65,66]. Among various scenarios of how this can be achieved, two important mechanisms are associated with MDSC and FoxP3^+^ T cells [71,72]. In the same figure, for comparison, we also plot the output from the deterministic model in Equation (Equation 1), which shows decaying oscillations around the deterministically stable steady state that are qualitatively similar to those observed in experimental data. These experimental and theoretical results indicate that taking averages across multiple experiments or simulations at each time point results in averaged T cell populations showing decaying oscillations, as observed in Figure 6. In contrast, a more detailed examination of individual experiments and single realisations (i.e., single numerical solutions) of the stochastic model in Equation (Equation 6), as illustrated in Figure 7, reveals substantial non-decaying oscillations in the magnitude of T cell populations around the mean value. The importance of this result lies in the fact that in laboratory or clinical settings, one observes only a single course of autoimmune disease, which may be characterised by sustained non-decaying oscillations.

It is important to note that the deterministic model in Equation (Equation 1) and its stochastic counterpart in Equation (Equation 6) can be scaled with respect to time, as well as magnitudes of cell populations. In other words, these models can exhibit the same qualitative behaviour, but on a different (ultimately, arbitrarily chosen) timescale by appropriately scaling the values of parameters, and the same applies to the values achieved by state variables. To give an example, if one were to replace state variables $T_{in}$, $T_{reg}$ and $T_{aut}$ by ${\hat{T}}_{in}=kT_{in}$, ${\hat{T}}_{reg}=kT_{reg}$ and ${\hat{T}}_{aut}=kT_{aut}$, respectively, where *k* is some positive scaling factor, while simultaneously replacing parameters $\mu_{a}$, $\lambda_{in}$, $\lambda_{r}$, σ and δ by ${\hat{\mu }}_{a}=\mu_{a}/k$, ${\hat{\lambda }}_{in}=k\lambda_{in}$, ${\hat{\lambda }}_{r}=k\lambda_{r}$, $\hat{\sigma }=\sigma /k$, and $\hat{\delta }=\delta /k$, respectively, this would result in the modified system having exactly the same dynamics as the original system, except that populations of T cells would now be scaled with a factor *k*. Interestingly, the rates $\rho_{1}$ and $\rho_{2}$ characterising the influence of cytokine growth factor on proliferation of different types of T cells, as well as death/clearance rates $d_{in}$, $d_{r}$, $d_{a}$ and $d_{i}$ remain unchanged.

The importance of having such scalings that keep qualitative behaviour but change the magnitudes to state variables, or alternatively, modify the timescales of the processes, lies in the fact that it allows one to much better fit the output of the model to experimental data, and this is particularly important in cases where only some, or a small number of, parameters can be properly measured experimentally.

## 4. Discussion

In this study, we analysed the role of stochasticity in autoimmune dynamics, paying particular attention to the emergence of sustained stochastic oscillations in cell populations in individual realisations of the model, and showed how the variance of these stochastic oscillations depends on model parameters. Immunologically, these stochastic oscillations correspond to an autoimmune process, which can develop in the model when regulatory T cells do not sufficiently suppress autoreactive T cells. We discovered that cell populations oscillate more substantially (i.e., have a larger variance) when cytokine growth factors are more strongly enhancing the proliferation of autoreactive T cells, or when they are less strongly enhancing the proliferation of regulatory T cells. We also compared numerical results from the model with experimental measurements of T cell populations during progression of EAU in B10.RIII and C57BL/6 mice, and the qualitative agreement is excellent, suggesting that the model is able to capture essential aspects of the immune response dynamics. In the specific context of autoimmunity, stochasticity is known to play a number of important roles, controlling the balance of activation and inhibition of T cell regulation [73] and expression of autoantigens and peripheral tissue antigens [74,75], as well as many other gene regulatory events associated with immune response [76]. A very recent review by Macfarlane et al. [77] has specifically highlighted an extremely important role played by stochasticity in triggering and mediating the progress of rheumatoid arthritis, a chronic autoimmune condition characterised by inflammation of joints, and suggested that stochastic models need to be developed in order to better understand the dynamics of this disease, and to optimise its management and potential treatments. In application to EAU dynamics, stochasticity manifests itself in the fact that, even though experiments are performed on genetically identical mice with exactly the same immunisation protocol, there is clinically significant variation in the time course of autoimmune disease between individual eyes. Such variation can only be attributed to stochastic factors, and this is exactly what we have explored with our model.

Whereas mathematical models presented in this paper are able to qualitatively reproduce dynamical behaviours observed in experiments both for averages, and for individual stochastic realisations, they do not currently capture the initial slow prodromal phase of the developing immune response prior to the first major peak in the number of T cells. This is an interesting and important challenge from the perspective of mathematical modelling, which has appeared in the context of modelling the dynamics of immune responses to viral infections [78,79,80], as well as in the studies of early stages of outbreaks of infectious diseases modelled at population level [81,82]. There are several methods for addressing this problem, such as including time delays or additional compartments to better represent the initial phase of the immune response [83,84,85,86], or considering spatial aspects of the immune dynamics using such approaches as partial differential equations, cellular automata, or multi-scale methods [17,87,88].

There are other research directions in which the work presented in this paper could be extended. While we focused on the analysis of stochastic effects in autoimmune dynamics that are associated with fluctuations in the numbers of different cell populations for fixed values of parameters, it may also be instructive to investigate the effects of time-dependent and/or stochastically varied parameters. This is known as “environmental stochasticity" in ecological [89,90,91] and epidemiological models [92,93], and it can be represented mathematically either by replacing constant parameters by time-dependent random functions, or by adding randomly fluctuating terms directly to deterministic equations. Random variation in parameters could be argued to provide a more realistic representation of behaviour of the immune system during a complex multi-factor process of immune response, as well as during onset and progression of autoimmune disease.

From an experimental perspective, an interesting and important question is whether the autoimmune disease develops in the same manner in different organs of the body. In this respect, eyes present particularly interesting organs, because they are known to enjoy what is known as “ocular immune privilege” [94,95,96,97], which means that eyes are relatively isolated from the rest of the immune system by a blood-ocular barrier. Ocular immune privilege is manifested by a very specialised microenvironment in the eye, characterised by the presence of immunosuppressive cells and anti-inflammatory cytokines. This environment resists the influx of autoreactive T cells not eliminated through a thymic selection, and each eye represents a single realisation of the disease process. Within the framework of the model presented in this paper, an important open question is whether there is statistically significant variation in the dynamics of autoimmune disease developing in different eyes of the same mouse, or between the eyes of different but genetically identical mice, all inoculated in the manner described in Section 2.

Another interesting problem directly related to ocular immune privilege concerns *sympathetic ophthalmia* [98,99], where a trauma in one eye results in the generation of eye antigens that subsequently cause an autoimmune disease in the non-damaged eye. Our earlier work based on deterministic models [34] has shed some light on pathogen-induced autoimmune dynamics associated with infection taking place in one organ of the body, and autoimmune reaction potentially occurring in another organ. Having in mind sympathetic ophthalmia as one of possible conditions falling under the same umbrella (although, most likely, not being directly associated with infectious triggers), it would be instructive to explore how such type of dynamics is affected by stochasticity, both in the numbers of cell populations, and in the values of parameters.

## Figures and Tables

**Figure 1 cells-09-00860-f001:**
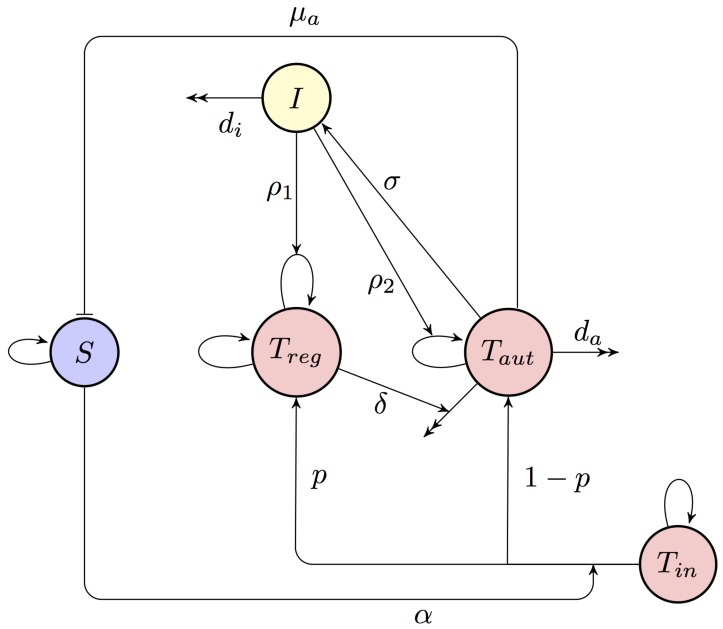
A diagram of the model of immune response. Blue circle indicates somatic cells that are the target of the autoimmune response, red circles denote different T cells (naïve, regulatory, and autoreactive T cells), and a yellow circle shows cytokine growth factor, such as IL-2.

**Figure 2 cells-09-00860-f002:**
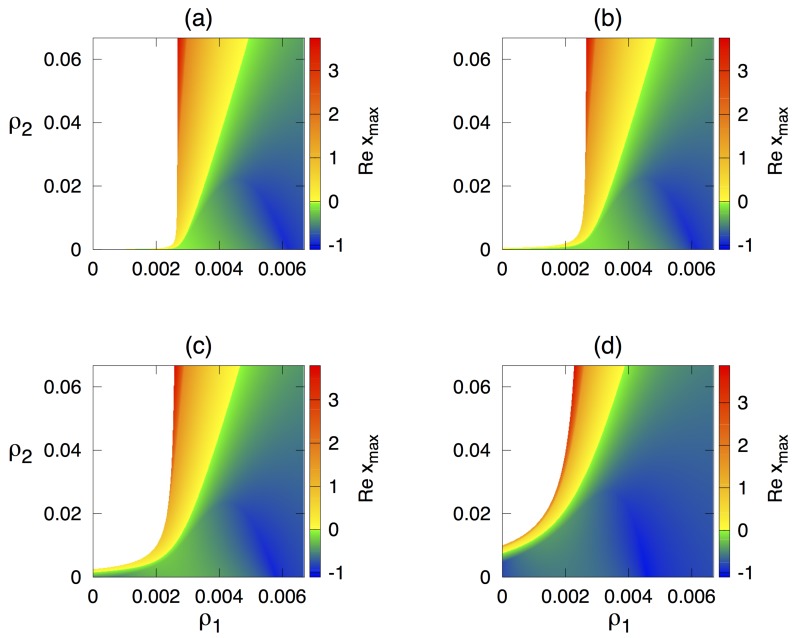
Stability of the steady state $E^{*}$ with parameter values from Table 1, and (**a**) $\delta =4.44\times 10^{-7}$; (**b**) $\delta =2.22\times 10^{-6}$; (**c**) $\delta =1.11\times 10^{-5}$; and (**d**) $\delta =4.44\times 10^{-5}$. The colour code shows max[Re(x)].

**Figure 3 cells-09-00860-f003:**
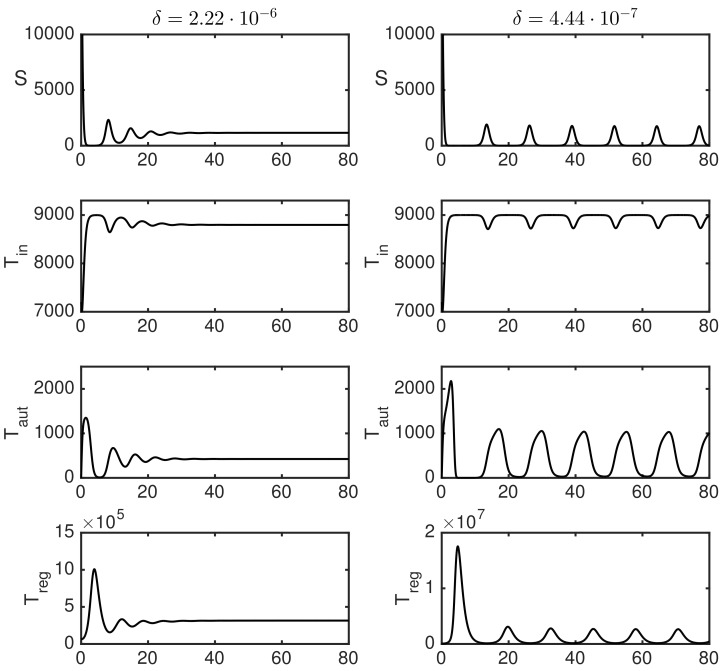
Numerical solution of the model in Equation (Equation 1) with parameter values from Table 1
$\rho_{1}=0.0022$, $\rho_{2}=4.4\times 10^{-4}$, and $\delta =2.22\times 10^{-6}$ (**left** column) or $4.44\times 10^{-7}$ (**right** column).

**Figure 4 cells-09-00860-f004:**
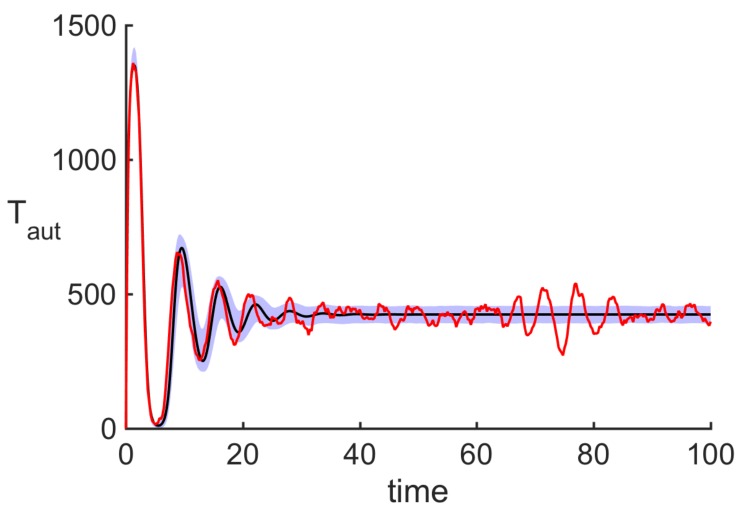
The black curve shows deterministic trajectory satisfying the macroscopic system in Equation (Equation 1), the red curve illustrates a single stochastic realisation of the model in Equation (Equation 6) and the purple region indicates the area of one standard deviation from the mean trajectory at each moment of time, as computed over 20,000 stochastic realisations. Parameter values are the same as in Figure 3, with $\delta =2.22\cdot 10^{-6}$.

**Figure 5 cells-09-00860-f005:**
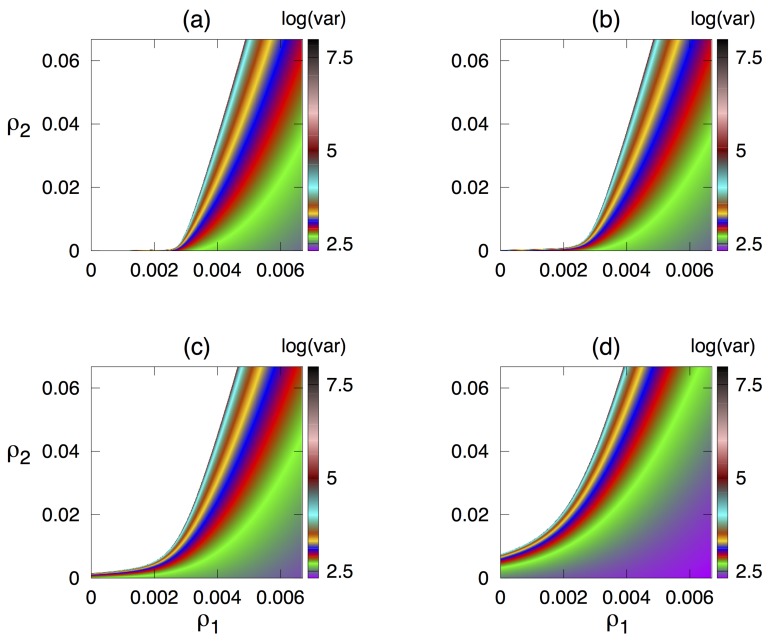
Variance in the number of autoreactive T cells during stochastic oscillations around deterministically stable steady state $E^{*}$ depending on parameters $\rho_{1}$ and $\rho_{2}$, with other parameter values as in Figure 4 and (**a**) $\delta =4.44\times 10^{-7}$; (**b**) $\delta =2.22\times 10^{-6}$; (**c**) $\delta =1.11\times 10^{-5}$; and (**d**) $\delta =4.44\times 10^{-5}$.

**Figure 6 cells-09-00860-f006:**
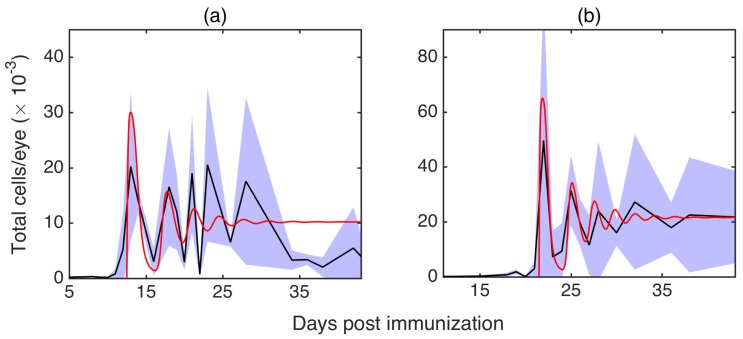
Intra-retinal numbers of CD4^+^ cells in B10.RIII mice (**a**) and CL57/B6 mice (**b**) at multiple time points following immunisation with RBP-3 peptides, with 10 realisations for B10.RIII mice and 8 realisations for CL57/B6 mice. Black curves indicate averages, shaded regions show areas of one standard deviation from the mean and red curves are numerical solutions of the model in Equation (Equation 1).

**Figure 7 cells-09-00860-f007:**
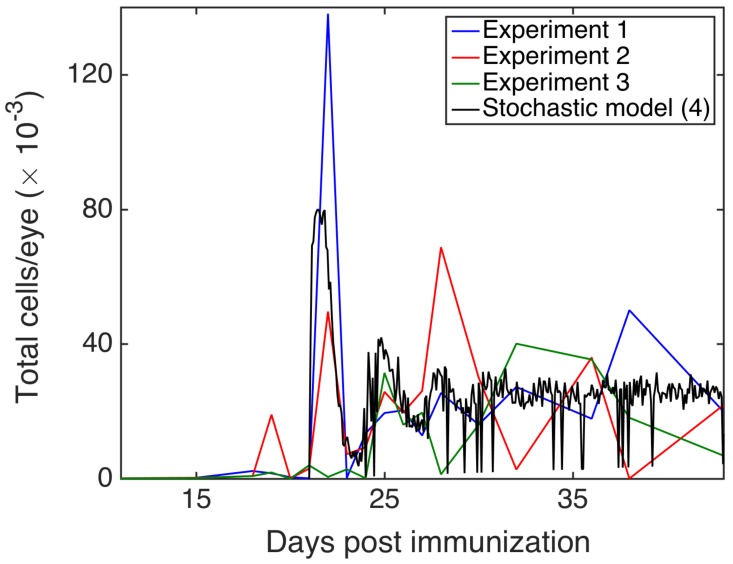
Experimental values of CD4^+^ cells from three experiments in CL57/B6 mice, compared to a single realisation of the stochastic model in Equation (Equation 6).

**Table 1 cells-09-00860-t001:** Table of parameters.

Parameter	Value	Parameter	Value
*r*	2	*N*	20,000
$\mu_{a}$	0.0044	α	$4\times 10^{-5}$
$\lambda_{in}$	18,000	$d_{in}$	2
$\lambda_{r}$	54,000	$d_{r}$	0.8
*p*	0.4	δ	$2.22\times 10^{-6}$
σ	0.8	$d_{i}$	0.6

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
