# Peer review of "Quantifying the Role of Stochasticity in the Development of Autoimmune Disease"

_cells, 2020, doi:10.3390/cells9040860_

Round 1

Reviewer 1 Report

The immune system is a dynamic network involved in multi-level biological processing. Through advanced clinical or laboratory technologies, we have greatly got knowledge’s on how immune system operated to facing self or non-self stimulus. However, the mechanism regarding how these individual parts systemically coordinated or act locally remains unrevealed. This manuscript from Dr. Nicholson et al described a mathematical model to test the role of stochasticity in the development of the autoimmune disease by testing T cell population with two published experimental autoimmune uveoretinitis (EAU) studies. Overall, the whole study has low scientific significance and poor novelty. Major suggestion include:

The same group has published a similar model and similar studies on stochastic effects in autoimmune and infection immunity. It is important to compare the difference of the roles of stochasticity in different immune response condition, and clearly describe the “exact” meaning of stochastic effects in autoimmune disease, especially EAU. For example, how stochasticity affects the development of autoimmune disease? What is the meaning of stochasticity for understanding the mechanism of autoimmune disease? Unbiased data to test the hypothesis model is critical. However, this study only analysis two groups of study by a single measurement, T cells, which is not convincing. More experimental analysis will be appreciated. Luckily, the mathematic modeling so far is not the mainstream for immunology research. To consider the readers of CELLS journal, the discussion section needs to be reorganized in plain langue for non-math readers.

Author Response

We are grateful to the Reviewer for careful reading of our manuscript and for their helpful comments. We would like to note that our manuscript is an Invited Paper to a Special Issue of “Cells” on “Quantitative models of autoimmunity”, so addresses modelling and, we feel, fits well within the scope of this special issue, which is dedicated specifically to mathematical models of various aspects of autoimmunity.

In terms of novelty, there is a major distinction between this work and our earlier publications on mathematical models of autoimmunity. In our earlier models, the emphasis was always on understanding the dynamics of T cells with tunable activation thresholds, and on autoimmunity arising through a breakdown of immune tolerance as a byproduct of immune response against an infection. In contrast, in this manuscript, we consider what is effectively a closed system of interacting cells in a particular organ of the body in the absence of any exogenous infections, and our model provides important insights into how sustained stochastic oscillations can arise in such a situation, and what their variance would be. Also, all our earlier models have been conceptual/theoretical, and this is the first time when we are also matching the results of our model to two sets of experiments, which by itself is quite rare in immunology, where mostly there are either mathematical models, or experiments, as the Reviewer has also noted.

In light of the fact that all available experimental evidence shows that T cells are necessary and sufficient to cause EAU, we have focused specifically on data describing their population biology in our experimental measurements. This aligns our model with the current understanding of immunology of autoimmune disease (and a substantial experimental experience of our lab in this field), and we have added a comment about this on p. 6.

We are grateful to the Reviewer for drawing our attention to the need to elucidate the role of stochasticity in autoimmune dynamics in general, and more specifically, in EAU. To address this issue, we have added an extended discussion of this point on pp. 11-12 together with a number of recent references, and have also reformulated our results from a less mathematical and more biological perspective to make it clearer to the audience of Cells journal, as suggested by the Reviewer.

Reviewer 2 Report

The manuscript entitled "Quantifying the role of stochasticity in the development of autoimmune disease" by  Lindsay B. Nicholson et. al is a well-written article. The authors have provided a sufficient introduction about previously existing models and clearly explained the need for the proposed model. In this paper, the authors addressed issues like the variance of stochastic fluctuations, their regularity and also validated them in experiments in mice model. I didn't have any major concerns regarding this study and recommend publishing this article in "Cells" with a minor revision.

Author Response

We are very grateful to the Reviewer for their positive and supportive assessment of our manuscript. We have done another careful proofreading of the manuscript and corrected a number of minor typos.